# Local Bayesian optimization via maximizing probability of descent

**Quan Nguyen**[*1]    **Kaiwen Wu**[*2]    **Jacob R. Gardner**[2]    **Roman Garnett**[1]

[1]Washington University in St. Louis    [2]University of Pennsylvania

{quan,garnett}@wustl.edu
{kaiwenwu,jacobrg}@seas.upenn.edu

## Abstract

Local optimization presents a promising approach to expensive, high-dimensional black-box optimization by sidestepping the need to globally explore the search space. For objective functions whose gradient cannot be evaluated directly, Bayesian optimization offers one solution – we construct a probabilistic model of the objective, design a policy to learn about the gradient at the current location, and use the resulting information to navigate the objective landscape. Previous work has realized this scheme by minimizing the variance in the estimate of the gradient, then moving in the direction of the expected gradient. In this paper, we re-examine and refine this approach. We demonstrate that, surprisingly, the expected value of the gradient is not always the direction maximizing the probability of descent, and in fact, these directions may be nearly orthogonal. This observation then inspires an elegant optimization scheme seeking to maximize the probability of descent while moving in the direction of most-probable descent. Experiments on both synthetic and real-world objectives show that our method outperforms previous realizations of this optimization scheme and is competitive against other, significantly more complicated baselines.

## 1 Introduction

The optimization of expensive-to-evaluate, high-dimensional black-box functions is ubiquitous in machine learning, science, engineering, and beyond; examples range from hyperparameter tuning [21] and policy search in reinforcement learning [3, 7], to configuring physics simulations [14]. High-dimensional global optimization faces an inherent difficulty stemming from the curse of dimensionality, as a thorough exploration of the search space becomes exponentially more expensive. It is more feasible to seek to *locally* optimize these high-dimensional objective functions, as we can then sidestep this inherent burden. This is true even in settings where we cannot directly observe the gradient of the objective function, as we may appeal to sophisticated techniques such as Bayesian optimization to nonetheless learn about the gradient of the objective through noisy observations, and then use this knowledge to navigate the high-dimensional search space locally.

A realization of this scheme has been proposed by Müller et al. [18], where a Gaussian process (GP) is used to model the objective function, and observations are designed to alternate between minimizing the variance – and thus uncertainty – of the GP's estimate of the gradient of the objective at a given location, then moving in the direction of the expected gradient. Although this approach seems natural, it fails to account for some nuances in the distribution of the directional derivative induced by the GP. Specifically, it turns out that beliefs about the gradient with *identical* uncertainty may nonetheless have *different* probabilities of descent along the expected gradient. Further and perhaps surprisingly, the expected gradient is not necessarily the direction maximizing the probability of descent – in fact, these

---

[*]Equal contribution.

36th Conference on Neural Information Processing Systems (NeurIPS 2022).

directions can be nearly orthogonal. In other words, simply minimizing the gradient variance and moving in the direction of the expected gradient may lead to suboptimal (local) optimization performance.

With this insight, we propose a scheme for local Bayesian optimization that alternates between identifying the direction of most probable descent, then moving in that direction. The result is a local optimizer that is efficient by design. To this end, we derive a closed-form solution for the direction of most probable descent at a given location in the input space under a GP belief about the objective function. We then design a corresponding closed-form acquisition function that optimizes (an upper bound of) the one-step maximum descent probability. Taken together, these components comprise an elegant and efficient optimization scheme. We demonstrate empirically that, across many synthetic and real-world functions, our method outperforms the aforementioned prior realization of this framework and is competitive against other, significantly more complicated baselines.

## 2 Preliminaries

We first introduce the problem setting and the local Bayesian optimization framework. We aim to numerically solve optimization problems of the form:

$$\text{given } \mathbf{x}_0 \in D, \text{ find } \mathbf{x}^* = \arg\min_{\mathbf{x} \in D(\mathbf{x}_0)} f(\mathbf{x}),$$

where $f \colon D \to \mathbb{R}$ is the black-box objective function we wish to optimize locally from a starting point $\mathbf{x}_0$, and $D(\mathbf{x}_0)$ is the local region around $\mathbf{x}_0$ inside the domain $D$. We model the objective function as a black box, and only assume that we may obtain potentially noisy function evaluations $y = f(\mathbf{x}) + \varepsilon$, where $\varepsilon \sim \mathcal{N}(0, \sigma^2)$, at locations of our choosing. We further assume the gradient cannot be measured directly, but only estimated from such noisy evaluations of the function. Finally, we consider the case where querying the objective is relatively expensive, limiting the number of times it may be evaluated. This constraint on our querying budget requires strategically selecting where to evaluate during optimization.

Bayesian optimization (BO) is one potential approach to this problem that offers unparalleled sample efficiency. BO constructs a probabilistic model of the objective function, typically a Gaussian process (GP) [19], and uses this model to design the next point(s) to evaluate the objective. After each observation, the GP is updated to reflect our current belief about the objective, which is then used to inform future decisions. We refer the reader to Garnett [6] for a thorough treatment of GPs and BO.

### 2.1 Local Bayesian optimization

In many applications, the objective function $f$ is high-dimensional. The curse of dimensionality poses a challenge for BO, as it will take exponentially more function evaluations to sufficiently cover the search space and find the global optimum. It may be more fruitful, therefore, to instead pursue *local* optimization, where we aim to descend from the current location, by probing the objective function in nearby regions to learn about its gradient.

It turns out the BO framework is particularly amenable to this idea, as a GP belief on the objective function induces a *joint* GP belief with its gradient [19], which we may use to guide local optimization. In particular, given a GP belief about the objective function $f$ with a once-differentiable mean function $\mu$ and a twice-differentiable covariance function $K$, the joint distribution of noisy function evaluations observations $(\mathbf{X}, \mathbf{y})$ and the gradient of $f$ at some point $\mathbf{x}$ is

$$p\left(\begin{bmatrix} \mathbf{y} \\ \nabla f(\mathbf{x}) \end{bmatrix}\right) = \mathcal{N}\left(\begin{bmatrix} \mu(\mathbf{X}) \\ \nabla \mu(\mathbf{x}) \end{bmatrix}, \begin{bmatrix} K(\mathbf{X}, \mathbf{X}) + \sigma^2 \mathbf{I} & K(\mathbf{X}, \mathbf{x})\nabla^\top \\ \nabla K(\mathbf{x}, \mathbf{X}) & \nabla K(\mathbf{x}, \mathbf{x})\nabla^\top \end{bmatrix}\right).$$

Here, when placed in front of $K$, the differential operator $\nabla$ indicates that we are taking the derivative of $K$ with respect to its first input; when placed behind $K$, it indicates the derivative is with respect to its second input. Conditioned on the observations $(\mathbf{X}, \mathbf{y})$, the posterior distribution of the derivative $\nabla f(\mathbf{x})$ may be obtained as:

$$p\big(\nabla f(\mathbf{x}) \mid \mathbf{x}, \mathbf{X}, \mathbf{y}\big) = \mathcal{N}(\boldsymbol{\mu}_\mathbf{x}, \Sigma_\mathbf{x}),$$

$$\text{where } \boldsymbol{\mu}_\mathbf{x} = \nabla \mu(\mathbf{x}) + \nabla K(\mathbf{x}, \mathbf{X})\big(K(\mathbf{X}, \mathbf{X}) + \sigma^2 \mathbf{I}\big)^{-1}\big(\mathbf{y} - \mu(\mathbf{X})\big), \tag{1}$$

$$\Sigma_\mathbf{x} = \nabla K(\mathbf{x}, \mathbf{x})\nabla^\top - \nabla K(\mathbf{x}, \mathbf{X})\big(K(\mathbf{X}, \mathbf{X}) + \sigma^2 \mathbf{I}\big)^{-1} K(\mathbf{X}, \mathbf{x})\nabla^\top.$$

Given the ability to reason about the objective function gradient given noisy function observations, we may realize a Bayesian local optimization scheme as follows. From a current location $\mathbf{x}$, we devise a policy that first designs observations seeking relevant information about the gradient $\nabla f(\mathbf{x})$, then, once satisfied, moves within the search space to a new location (that is, update $\mathbf{x}$) seeking to descend on the objective. A particular realization of this local BO scheme named GIBO was investigated by Müller et al. [18]. In that study, the authors choose to learn about $\nabla f(\mathbf{x})$ by minimizing the uncertainty (quantified by the trace of the posterior covariance matrix) about the gradient, followed by moving in the direction of the expected gradient. This algorithm may be thought of as simulating gradient descent, as it actively builds then follows a noisy estimate of the gradient. Although effective, GIBO fails to account for nuances in our belief about the objective function gradient and may behave suboptimally during optimization as a result. Our work addresses this gap by exploiting the rich structure in the belief about $\nabla f(\mathbf{x})$ to design an elegant and principled policy for local BO.

## 2.2 Related work

We re-examine and extend the work of Müller et al. [18], who proposed using local BO for the purpose of policy search in reinforcement learning (RL). As mentioned, their proposed algorithm GIBO alternates between minimizing the variance of the estimate of the gradient – this is analogous to the goal of A-optimality in optimal design – and moving in the direction of the expected gradient. This scheme was shown to outperform baselines such as global BO using expected improvement [11] and the evolutionary algorithm CMA-ES [8] on several problems. Prior to this work, Mania et al. [16] noted that local black-box optimization is a promising approach for RL. They developed a simple algorithm, Augmented Random Search (ARS), that estimates the gradient of the objective via finite differencing and random perturbations; this simple method was competitive in their experiments on RL tasks. GIBO and ARS are the two main baselines that we will be comparing our method against.

As mentioned, scaling to high-dimensional problems has been an enduring challenge in the BO community, and there have been many proposals to make BO "more local" as a way to relieve the burden of the curse of dimensionality. In particular, several lines of research have proposed restricting the search space to only specific regions, e.g., maintaining a belief about the local optimum [1], using trust regions [5, 25], and forcing queries to stay close to past observations [13]. Among these, of note is the TuRBO algorithm [5], which expands and shrinks the size of its trust regions based on the optimization history within each region, and has been shown to achieve strong performance across many tasks. We include TuRBO as another baseline in our experiments.

Other approaches have considered dynamically switching from global and gradient-based local optimization, particularly when a local region is believed to contain the global optimum. For example, McLeod et al. [17] proposed alternating between global BO and using BFGS for local optimization when there is high certainty that we are close to the global optimum. Diouane et al. [4] leveraged the same scheme to identify good local regions and uses a trust region-based policy for its local phase. Wang et al. [26], on the other hand, proposed learning about which subregions of the search space are more likely to contain good objective values and should be locally exploited using Monte Carlo tree search, by recursively partitioning the space based on optimization performance. The authors also showed that when combined with TuRBO, their algorithm achieves state-of-the-art performance on a wide range of tasks. Our optimization method can replace the local optimizer in these approaches, and in general can act as a subroutine within a larger framework relying on local optimization.

Tackling local optimization from a probabilistic angle, our method belongs to a larger class of probabilistic numerical methods; see chapter 4 of Hennig et al. [10] for a thorough discussion on probabilistic numerics for local optimization. Within this line of search are other efforts at leveraging probabilistic reasoning in optimization, including a Bayesian quasi-Newton algorithm that learns from noisy observations of the gradient [9], a probabilistic interpretation of the incremental proximal methods [2], and probabilistic line searches [15].

We note that Le Roux et al. [12] arrived at a similar update expression as our algorithm (see Sect. 3), though aiming at developing fast optimization algorithms for good generalization, a different problem from BO. Moreover, their derivation is devoted to justifying the natural gradient descent. In particular, they show that the descent direction maximizing the probability of not increasing generalization error is precisely the natural gradient direction.

# 3 Maximizing probability of descent

What behavior is desirable for a local optimization routine that values sample efficiency? We argue that we should seek to quickly identify directions that will, *with high probability,* yield progress on the objective function. Pursuing this idea requires reasoning about the probability that a given direction leads "downhill" from a given location. Although one might guess that the direction most likely to lead downhill is always the (negative) expected gradient, this is not necessarily the case.

Consider the directional derivative of the objective $f$ with respect to a unit vector $\mathbf{v}$ at point $\mathbf{x}$:

$$\nabla_{\mathbf{v}} f(\mathbf{x}) = \mathbf{v}^\top \nabla f(\mathbf{x}),$$

which quantifies the rate of change of $f$ at $\mathbf{x}$ along the direction of $\mathbf{v}$. According to our GP belief, $\nabla f(\mathbf{x})$ follows a multivariate normal distribution, so the directional derivative $\nabla_{\mathbf{v}} f(\mathbf{x})$ is then:

$$p\big(\nabla_{\mathbf{v}} f(\mathbf{x}) \mid \mathbf{x}, \mathbf{v}\big) = \mathcal{N}\big(\mathbf{v}^\top \boldsymbol{\mu}_{\mathbf{x}}, \mathbf{v}^\top \Sigma_{\mathbf{x}} \mathbf{v}\big),$$

where $\boldsymbol{\mu}_{\mathbf{x}}$ and $\Sigma_{\mathbf{x}}$ are the mean and covariance matrix of the normal belief about $\nabla f(\mathbf{x})$, as defined in Eq. (1). This distribution allows us to reason about the probability that we descend on the objective function by moving along the direction of $\mathbf{v}$ from $\mathbf{x}$, which is simply the probability that the directional derivative is negative. Thus, we have the following definition.

**Definition 3.1** (Descent probability and most probable descent direction)**.** *Given a unit vector $\mathbf{v}$, the descent probability of the direction $\mathbf{v}$ at the location $\mathbf{x}$ is given by*

$$\Pr\big(\nabla_{\mathbf{v}} f(\mathbf{x}) < 0 \mid \mathbf{x}, \mathbf{v}\big) = \Phi\left(-\frac{\mathbf{v}^\top \boldsymbol{\mu}_{\mathbf{x}}}{\sqrt{\mathbf{v}^\top \Sigma_{\mathbf{x}} \mathbf{v}}}\right), \tag{2}$$

*where $\Phi$ is the CDF of the standard normal distribution. If $\mathbf{v}^*$ achieves the maximum descent probability $\mathbf{v}^* \in \arg\max_{\mathbf{v}} \Pr\big(\nabla_{\mathbf{v}} f(\mathbf{x}) < 0 \mid \mathbf{x}, \mathbf{v}\big)$, then we call $\mathbf{v}^*$ a most probable descent direction.*

Note that the definition Eq. (2) is scaling invariant. Thus, the length of $\mathbf{v}^*$ does not matter since the descent probability only depends on its direction. Moreover, we note that descent probability depends on both the expected gradient $\boldsymbol{\mu}_{\mathbf{x}}$ and the gradient uncertainty $\Sigma_{\mathbf{x}}$. Therefore, learning about the gradient by minimizing uncertainty via the trace of the posterior covariance matrix (which does not consider the expected gradient) and moving in the direction of the negative expected gradient (which does not consider uncertainty in the gradient) in a decoupled manner may lead to suboptimal behavior. We first present a simple example to demonstrate the nuances that are not captured by this scheme and to motivate our proposed solution.

## 3.1 The (negative) expected gradient does not always maximize descent probability

In Fig. 1, we show polar plots of the descent probability $\Pr\big(\nabla_{\mathbf{v}} f(\mathbf{x}) < 0 \mid \mathbf{x}, \mathbf{v}\big)$ with respect to different beliefs about the gradient. The angles in the polar plots are the angles between $\mathbf{v}$ and the vector $[1, 0]^\top$. Critically for the discussion below, the uncertainty in the gradient, as measured by the trace of the covariance matrix, is identical for all three examples.

In the first example in the left panel of Fig. 1, the negative expected gradient happens to maximize the descent probability, and moving in this direction is almost certain to lead downhill. In the middle panel, the expected gradient is the same as in the left panel, but the covariance matrix has been permuted. Here, the negative expected gradient again maximizes the descent probability; however, the largest descent probability is now much lower. In fact, there is non-negligible probability that the descent direction is in the *opposite* direction. This is because most of the uncertainty we have about the gradient concentrates on the first element of $\boldsymbol{\mu}_{\mathbf{x}}$, which determines its direction. We note that the situation in the left panel is inarguably preferable to that in the middle panel, but distinguishing these two is impossible from uncertainty in $\nabla f(\mathbf{x})$ alone.

Finally, in the right panel, the direction of the expected gradient has rotated with respect to that in the first two panels. Now the (negative) expected gradient is entirely different from the most probable descent direction. Intuitively, the variance in the first coordinate is much smaller than in the second coordinate, and thus the mean in the first coordinate is more likely to have the same sign as the true gradient. However, using negative expected gradient as a descent direction entirely ignores the uncertainty estimate in the gradient. This example shows that, when we reason about the descent of a function, the mean vector $\boldsymbol{\mu}_{\mathbf{x}}$ and the covariance matrix $\Sigma_{\mathbf{x}}$ need to be jointly considered, as the probability of descent depends on both of these quantities (Eq. (2)).

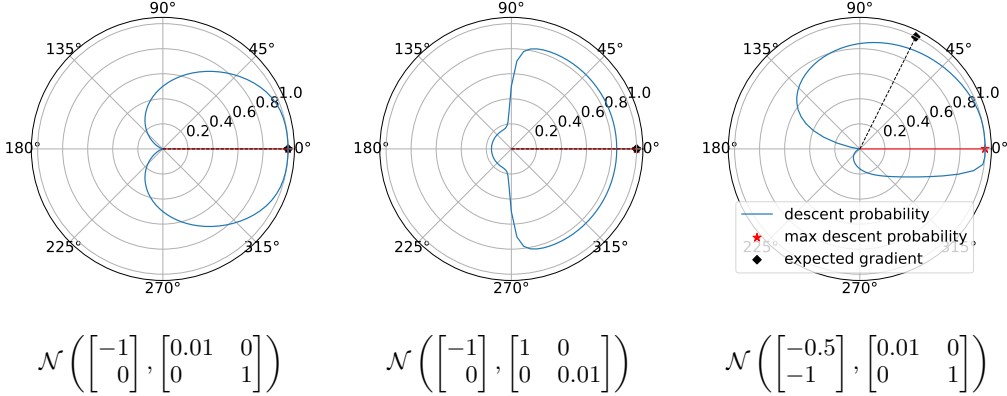

Figure 1: Polar plots of descent probability (blue). The most probable descent direction $\mathbf{v}^*$ is marked in red. The direction of the (negative) expected gradient is marked in **black**. **Left:** the direction $\mathbf{v}^*$ and the negative expected gradient match exactly. **Center:** given the same level of uncertainty, the maximum descent probability has reduced from near certainty to only $84\%$. **Right:** the expected gradient does not maximize the descent probability. See Sect. 3.1 for discussion.

## 3.2 Computing the most probable descent direction

In light of the above discussion, we propose a local BO algorithm centered entirely around the local descent probability. As a first step, we show in the following how to compute the most probable descent direction $\mathbf{v}^* = \arg\max_{\mathbf{v}} \Pr\left(\nabla_{\mathbf{v}} f(\mathbf{x}) < 0 \mid \mathbf{x}, \mathbf{v}\right)$ at a given location given data.

**Theorem 3.1.** *Suppose that the belief about the gradient is $p\left(\nabla f(\mathbf{x}) \mid \mathbf{x}, \mathbf{X}, \mathbf{y}\right) = \mathcal{N}(\boldsymbol{\mu}_{\mathbf{x}}, \Sigma_{\mathbf{x}})$, where the posterior covariance $\Sigma_{\mathbf{x}}$ is positive definite. Then, the unique (up to scaling) most probable descent direction is*

$$\arg\max_{\mathbf{v}} \Pr\left(\nabla_{\mathbf{v}} f(\mathbf{x}) < 0 \mid \mathbf{x}, \mathbf{v}\right) = -\Sigma_{\mathbf{x}}^{-1} \boldsymbol{\mu}_{\mathbf{x}}$$

*with the corresponding maximum descent probability*

$$\max_{\mathbf{v}} \Pr\left(\nabla_{\mathbf{v}} f(\mathbf{x}) < 0 \mid \mathbf{x}, \mathbf{v}\right) = \Phi\left(\sqrt{\boldsymbol{\mu}_{\mathbf{x}}^{\top} \Sigma_{\mathbf{x}}^{-1} \boldsymbol{\mu}_{\mathbf{x}}}\right).$$

*Proof.* As $\Phi\left(\cdot\right)$ is monotonic, we can reframe the problem as

$$\mathbf{v}^* = \arg\max_{\mathbf{v}} \Pr\left(\nabla_{\mathbf{v}} f(\mathbf{x}) < 0 \mid \mathbf{x}, \mathbf{v}\right) = \arg\max_{\mathbf{v}} \Phi\left(-\frac{\mathbf{v}^{\top} \boldsymbol{\mu}_{\mathbf{x}}}{\sqrt{\mathbf{v}^{\top} \Sigma_{\mathbf{x}} \mathbf{v}}}\right) = \arg\max_{\mathbf{v}} -\frac{\mathbf{v}^{\top} \boldsymbol{\mu}_{\mathbf{x}}}{\sqrt{\mathbf{v}^{\top} \Sigma_{\mathbf{x}} \mathbf{v}}}.$$

Next, we square the objective, and the maximizer is still the same (up to sign). That is, if $\mathbf{v}^*$ is the maximizer of the squared objective:

$$\mathbf{v}^* = \arg\max_{\mathbf{v}} \frac{\mathbf{v}^{\top} \boldsymbol{\mu}_{\mathbf{x}} \boldsymbol{\mu}_{\mathbf{x}}^{\top} \mathbf{v}}{\mathbf{v}^{\top} \Sigma_{\mathbf{x}} \mathbf{v}}, \tag{3}$$

then either $\mathbf{v}^*$ or $-\mathbf{v}^*$ maximizes the descent probability. Let $\Sigma_{\mathbf{x}} = \mathbf{L}\mathbf{L}^{\top}$ be the Cholesky decomposition of $\Sigma_{\mathbf{x}}$, where $\mathbf{L}$ has to be nonsingular. A change of variable $\mathbf{v} = \mathbf{L}^{-\top} \mathbf{w}$ gives

$$\frac{\mathbf{v}^{\top} \boldsymbol{\mu}_{\mathbf{x}} \boldsymbol{\mu}_{\mathbf{x}}^{\top} \mathbf{v}}{\mathbf{v}^{\top} \Sigma_{\mathbf{x}} \mathbf{v}} = \frac{\mathbf{w}^{\top} \mathbf{L}^{-1} \boldsymbol{\mu}_{\mathbf{x}} \boldsymbol{\mu}_{\mathbf{x}}^{\top} \mathbf{L}^{-\top} \mathbf{w}}{\mathbf{w}^{\top} \mathbf{w}},$$

which is exactly the Rayleigh quotient of $\mathbf{L}^{-1} \boldsymbol{\mu}_{\mathbf{x}} \boldsymbol{\mu}_{\mathbf{x}}^{\top} \mathbf{L}^{-\top}$. Note that this is a rank-1 matrix with top eigenvector $\mathbf{L}^{-1} \boldsymbol{\mu}_{\mathbf{x}}$ and corresponding eigenvalue $\boldsymbol{\mu}_{\mathbf{x}}^{\top} \Sigma_{\mathbf{x}}^{-1} \boldsymbol{\mu}_{\mathbf{x}}$. Thus, the maximizer $\mathbf{w}^*$ is given by

$$\mathbf{w}^* = \mathbf{L}^{-1} \boldsymbol{\mu}_{\mathbf{x}}.$$

Therefore, the maximizer to Eq. (3) is $\mathbf{v}^* = \mathbf{L}^{-\top} \mathbf{L}^{-1} \mathbf{w}^* = \Sigma_{\mathbf{x}}^{-1} \boldsymbol{\mu}_{\mathbf{x}}$. Plug both $\Sigma_{\mathbf{x}}^{-1} \boldsymbol{\mu}_{\mathbf{x}}$ and $-\Sigma_{\mathbf{x}}^{-1} \boldsymbol{\mu}_{\mathbf{x}}$ back into Eq. (2). It is easy to check that the direction along $-\Sigma_{\mathbf{x}}^{-1} \boldsymbol{\mu}_{\mathbf{x}}$ is the desired maximizer. □

Theorem 3.1 states that the most probable descent direction can be computed by simply solving a linear system. Being able to compute this quantity allows us to always move within the search space in the direction that most likely improves the objective value, which, as we have seen, is not necessarily the negative expected gradient. This helps us to realize the "update" portion of our local BO algorithm, where we iteratively move from the current location $\mathbf{x}$ in the most probable descent direction $\mathbf{v}^*$. That is, we repeatedly update $\mathbf{x}$ with $\mathbf{x} + \delta\mathbf{v}^*$, where $\delta$ is a small constant that acts as a step size. This procedure is iterative in that we do not take one single step along a direction, but multiple small steps, always in the most probable descent direction at the current point, throughout. (Note that we do not observe the value of the objective function at any of these steps.)

It is important that we stop this iterative procedure when it becomes uncertain whether we can continue to descend. This is because we aim to move to a new location that decreases the value of the objective function, and thus should only move when descent is likely. A natural approach is to again use the maximum descent probability, which we can compute using Theorem 3.1. Specifically, we stop the iterative update when the maximum descent probability falls below a prespecified threshold $p_*$. Once we have stopped, the final updated $\mathbf{x}$ is the location we move to at the current iteration of the BO loop. In our experiments, we set the step size to $\delta = 0.001$ and the descent probability threshold to $p_* = 65\%$, which we find to work well empirically.

### 3.3 Acquisition function via look-ahead maximum descent probability

When the maximum descent probability falls below the threshold $p_*$, we begin selecting queries to learn about the gradient in the current location so as to maximize the probability of descent. Here we derive an acquisition function seeking data that will, in expectation, best improve the highest descent probability. For maximum flexibility, we consider the batch setting where we may gather multiple measurements simultaneously, although we only use the sequential case in our experiments.

In particular, the acquisition function we would like to use for a batch of potential query points $\mathbf{Z}$ is:

$$
\begin{aligned}
\alpha_0(\mathbf{Z}) &= \mathbb{E}_{\mathbf{y}|\mathbf{Z}}\left[\max_{\mathbf{v}} \Pr\left(\nabla_{\mathbf{v}} f(\mathbf{x}) < 0 \mid \mathbf{x}, \mathbf{Z}\right)\right] \\
&= \mathbb{E}_{\mathbf{y}|\mathbf{Z}}\left[\Phi\left(\sqrt{\boldsymbol{\mu}_{\mathbf{x}|\mathbf{Z}}^\top \Sigma_{\mathbf{x}|\mathbf{Z}}^{-1} \boldsymbol{\mu}_{\mathbf{x}|\mathbf{Z}}}\right)\right],
\end{aligned}
\tag{4}
$$

where $\boldsymbol{\mu}_{\mathbf{x}|\mathbf{Z}}$ and $\Sigma_{\mathbf{x}|\mathbf{Z}}$ are the posterior mean and covariance of the belief about $\nabla f(\mathbf{x})$, conditioned on a batch of observations at $\mathbf{Z}$ and a previously collected training set $(\mathbf{X}, \mathbf{y})$ which we have omitted for notational clarity. Note that the second equality is due to Theorem 3.1. The above acquisition function is exactly the look-ahead maximum descent probability. Namely, $\alpha_0(\mathbf{Z})$ is the expected maximum descent probability after querying $\mathbf{Z}$.

Unfortunately, this expectation is challenging to compute, so we opt for another acquisition function that approximates Eq. (4) via computing the expectation of an upper bound:

$$
\alpha(\mathbf{Z}) = \mathbb{E}_{\mathbf{y}|\mathbf{Z}}\left[\boldsymbol{\mu}_{\mathbf{x}|\mathbf{Z}}^\top \Sigma_{\mathbf{x}|\mathbf{Z}}^{-1} \boldsymbol{\mu}_{\mathbf{x}|\mathbf{Z}}\right].
\tag{5}
$$

We discard the (monotonic and concave) transformation given by the normal CDF and square root, thus optimizing an upper bound by Jensen's inequality. The advantage to this acquisition function $\alpha$ is that, remarkably, it has a closed-form expression, as we show below.

Note that $\boldsymbol{\mu}_{\mathbf{x}|\mathbf{Z}} = \boldsymbol{\mu}_{\mathbf{x}} + \Sigma_{\mathbf{x}\mathbf{Z}}\Sigma_{\mathbf{Z}}^{-1}(\mathbf{y}_{\mathbf{Z}} - \boldsymbol{\mu}_{\mathbf{Z}})$, where $\mathbf{y}_{\mathbf{Z}} \sim \mathcal{N}(\boldsymbol{\mu}_{\mathbf{Z}}, \Sigma_{\mathbf{Z}})$. Thus, the acquisition function in Eq. (5) is an expectation of a quadratic function over a Gaussian distribution. Let $\mathbf{L}\mathbf{L}^\top = \Sigma_{\mathbf{Z}}$ be the Cholesky decomposition of $\Sigma_{\mathbf{Z}}$ and denote $\mathbf{A} = \Sigma_{\mathbf{x}\mathbf{Z}}\mathbf{L}^{-\top}$. Then, the acquisition function can be written as an expectation over a standard normal $\boldsymbol{\zeta}$:

$$
\alpha(\mathbf{Z}) = \mathbb{E}_{\boldsymbol{\zeta} \sim \mathcal{N}(\mathbf{0}, \mathbf{I})}\left[(\boldsymbol{\mu}_{\mathbf{x}} + \mathbf{A}\boldsymbol{\zeta})^\top \Sigma_{\mathbf{x}|\mathbf{Z}}^{-1} (\boldsymbol{\mu}_{\mathbf{x}} + \mathbf{A}\boldsymbol{\zeta})\right].
$$

Expanding, we have:

$$
(\boldsymbol{\mu}_{\mathbf{x}} + \mathbf{A}\boldsymbol{\zeta})^\top \Sigma_{\mathbf{x}|\mathbf{Z}}^{-1} (\boldsymbol{\mu}_{\mathbf{x}} + \mathbf{A}\boldsymbol{\zeta}) = \boldsymbol{\mu}_{\mathbf{x}}^\top \Sigma_{\mathbf{x}|\mathbf{Z}}^{-1} \boldsymbol{\mu}_{\mathbf{x}} + 2\boldsymbol{\mu}_{\mathbf{x}}^\top \Sigma_{\mathbf{x}|\mathbf{Z}}^{-1} \mathbf{A}\boldsymbol{\zeta} + \boldsymbol{\zeta}^\top \mathbf{A}^\top \Sigma_{\mathbf{x}|\mathbf{Z}}^{-1} \mathbf{A}\boldsymbol{\zeta}.
$$

The expectation of each term can be computed in closed form. The first term is a constant and the second term vanishes. Finally, the third term is the expectation of a quadratic form, yielding:

$$
\alpha(\mathbf{Z}) = \boldsymbol{\mu}_{\mathbf{x}}^\top \Sigma_{\mathbf{x}|\mathbf{Z}}^{-1} \boldsymbol{\mu}_{\mathbf{x}} + \mathrm{tr}\left(\mathbf{A}^\top \Sigma_{\mathbf{x}|\mathbf{Z}}^{-1} \mathbf{A}\right).
$$

**Algorithm 1** Local BO via MPD

---

1: **inputs** starting location $\mathbf{x}$, number of iterations $N$, number of samples for learning the gradient $M$, step size $\delta$, and minimum descent probability threshold $p_*$.
2: Initialize the GP.
3: **for** $t = 0, \ldots, N$ **do**
4:     Observe the objective value: $y = f(\mathbf{x}) + \varepsilon$.
5:     Update the training data $\mathcal{D} \leftarrow \mathcal{D} \cup \{(\mathbf{x}, y)\}$ and retrain the GP.
6:     **for** $m = 1, \ldots, M$ **do**                         ▷ learning the gradient
7:         Query point: $\mathbf{z}^* = \arg\max_{\mathbf{z}} \alpha(\mathbf{z})$.
8:         Observe the objective value: $y_{\mathbf{z}} = f(\mathbf{z}) + \varepsilon$.
9:         Update the training data $\mathcal{D} \leftarrow \mathcal{D} \cup \{(\mathbf{z}, y_{\mathbf{z}})\}$ and the GP.
10:     **end for**
11:     **while** $\max_{\mathbf{v}} \Pr\left(\nabla_{\mathbf{v}} f(\mathbf{x}) < 0 \mid \mathbf{x}, \mathbf{v}\right) > p_*$ **do**   ▷ move by maximizing descent probability
12:         Compute the most probable descent direction $\mathbf{v}^* \leftarrow \arg\max_{\mathbf{v}} \Pr\left(\nabla_{\mathbf{v}} f(\mathbf{x}) < 0 \mid \mathbf{x}, \mathbf{v}\right)$.
13:         Move in the most probable descent direction: $\mathbf{x} \leftarrow \mathbf{x} + \delta\mathbf{v}^*$.
14:     **end while**
15: **end for**

---

This compact expression gives the closed-form solution to our acquisition function. Note that solving a linear system with respect to $\Sigma_{\mathbf{x}|\mathbf{Z}}$ can be performed efficiently using low-rank updates to the Cholesky decomposition of $\Sigma_{\mathbf{x}}$. Further, we may differentiate the acquisition function easily via automatic differentiation. This allows us to optimize the acquisition function trivially using any gradient-based optimizer such as L-BFGS with restart.

This completes our algorithm, local BO via most-probable descent, or MPD, which is summarized in Alg. 1. The algorithm alternates between learning about the gradient of the objective function using the acquisition function discussed above, and then iteratively moving in the most probable descent direction until further progress is unlikely, as described in Sect. 3.

## 4 Experiments

We now present results from extensive experiments that evaluate our method MPD against three baselines: (1) GIBO [18], which performs local BO by minimizing the trace of the posterior covariance matrix of the gradient and uses the expected gradient in the update step; (2) ARS [16], which estimates the gradient of the objective via finite difference with random perturbations; and (3) TuRBO [5], a trust region-based Bayesian optimization method.

Müller et al. [18] provide code implementation under the MIT license for GIBO, ARS, and various test objectives. We extend this codebase to implement MPD and conduct our own numerical experiments. For the synthetic (Sect. 4.1) and reinforcement learning (Sect. 4.2) objectives, we use the provided experimental settings. For the other objectives (Sect. 4.3), we set the number of samples to learn about the gradient per iteration $M = 1$. For each objective function tested, we run each algorithm ten times from the same set of starting points sampled from a Sobol sequence over the (box-bounded) domain. In each of the following plots, we show the progressive mean objective values as a function of the number of queries with error bars indicating (plus or minus) one standard error. Experiments were performed on a small cluster built from commodity hardware comprising approximately 200 Intel Xeon CPU cores (no GPUs), with approximately 10 GB of RAM available to each core. Our implementation is available at `https://github.com/kayween/local-bo-mpd`.

### 4.1 Synthetic objectives

Our first experiments involve maximizing, over the $d$-dimensional unit hypercube $[0, 1]^d$, synthetic objective functions that are generated by drawing samples from a GP with an RBF kernel. We refer to §4.1 of Müller et al. [18] for more details regarding the experimental setup. While Müller et al. [18] tested for dimensions up to 36, we opt for much higher-dimensional objectives: $d \in \{25, 50, 100\}$. Each run has a budget of 500 function evaluations. We visualize the results in Fig. 2, which shows that MPD was able to optimize these functions at a faster rate than the other baselines. Note that the difference in performance becomes larger as the dimensionality $d$ grows, indicating that our method scales well to high dimensions.

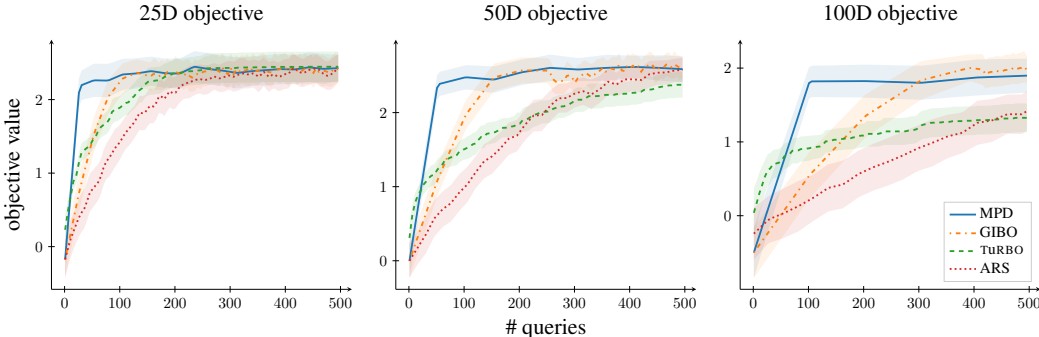

Figure 2: Progressive optimized objective value on high-dimensional synthetic functions. MPD consistently finds higher objective values faster than other baselines.

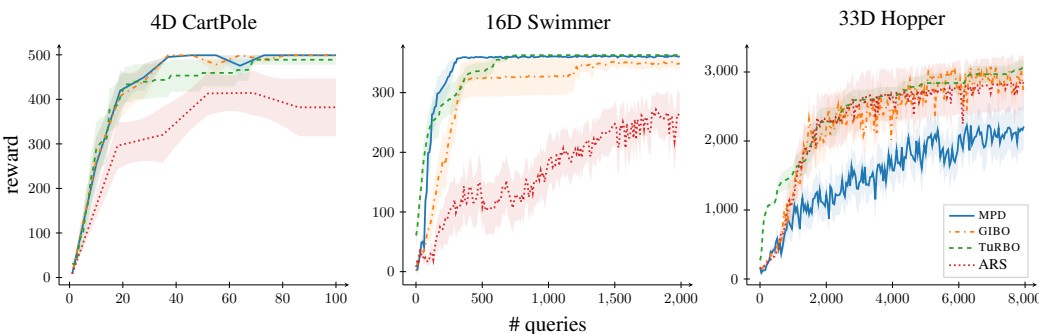

Figure 3: Progressive objective values observed on the MuJuCo tasks. MPD is competitive on CartPole and Swimmer.

## 4.2 MuJoCo objectives

The second set of experiments are reinforcement learning MuJoCo locomotion tasks [23], where each task involves learning a linear policy that maps states to actions to maximize the reward received from the learning environment. We use the same three environments in Müller et al. [18], CartPole-v1 with 4 parameters, Swimmer-v1 with 16, and Hopper-v1 with 33, to evaluate the methods and show the results in Fig. 3. MPD is competitive in the first two tasks but progresses slower than the other baselines on Hopper-v1. We conduct a thorough investigation into the cause of MPD's failure on the Hopper function and present our findings in Appx. B. In short, the experiments on Hopper-v1 employ a state normalization scheme (described in §3.3 of Müller et al. [18]) that leads to systematic differences in the behavior of GIBO and MPD. By controlling for the effect of state normalization in our comparison of the two algorithms, we find that the performance of GIBO and that of MPD are statistically comparable.

## 4.3 Other objective functions

We further evaluate our method on other real-world objective functions. The first two functions represent inverse problems from physics and engineering. The first is from electrical engineering, where we seek to maximize the fit of a theoretical physical model of an electronic circuit to observed data. There are nine parameters in total, and we set the budget to 500 evaluations. The second is a problem from cosmology [20], where we aim to configure a cosmological model/physical simulator to fit data observed from the Universe. In particular, our objective is to maximize the log likelihood of the physical model parameterized by various physics-related constants that are to be tuned. We follow the setting in Eriksson et al. [5], which presents a harder optimization problem with 12 parameters and much larger bounds, and set the budget at 2000 evaluations. Our third objective function uses the rover trajectory planning problem [27]. This involves tuning the locations of 100 points on a two-dimensional space that map the trajectory of a rover to minimize a cost, thus making up a 200-dimensional optimization problem. We set the budget to be 1000 function evaluations.

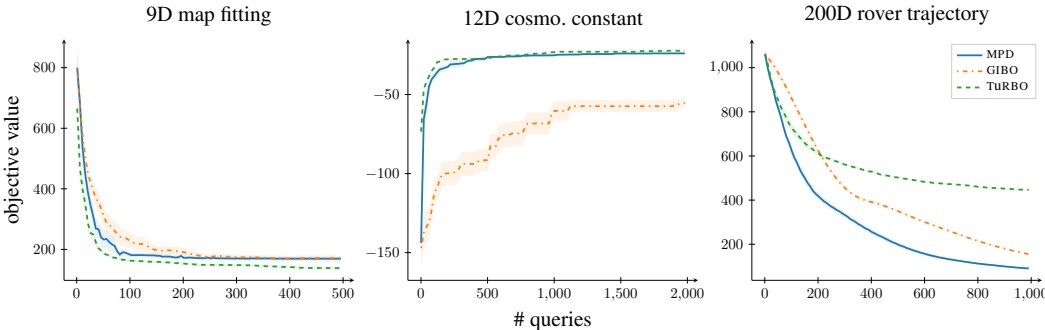

Figure 4: Progressive objective values observed on real-world tasks. MPD is competitive against other baselines on all tasks.

Table 1: Average terminal optimized objective values and standard errors of different variants of MPD. Results that are better than those of our baseline GIBO are highlighted **bold**.

| | 16D Swimmer (maximization) | 12D cosmo. constant (maximization) | 200D rover trajectory (minimization) |
|---|---|---|---|
| MPD$(p_* = 65\%, \delta = 10^{-3})$ | **360.50 (0.61)** | $-$**23.97 (0.34)** | **89.89 (3.88)** |
| GIBO | 348.88 (10.11) | $-55.25$ (3.23) | 152.77 (2.26) |
| trace + MPD | 350.58 (9.35) | $-$**27.72 (1.16)** | **84.17 (2.10)** |
| MPD + expected gradient | 340.12 (12.75) | $-$**21.24 (0.04)** | 293.08 (8.12) |
| MPD$(p_* = 50\%)$ | 342.36 (13.10) | $-$**24.29 (0.10)** | **51.48 (3.44)** |
| MPD$(p_* = 85\%)$ | 294.67 (38.16) | $-$**31.08 (0.86)** | **142.63 (5.57)** |
| MPD$(p_* = 95\%)$ | 15.97 (5.46) | $-$**31.86 (0.25)** | **140.44 (6.95)** |
| MPD$(\delta = 10^{-4})$ | **362.06 (0.63)** | $-$**24.22 (0.53)** | **90.99 (3.29)** |
| MPD$(\delta = 10^{-2})$ | 350.15 (10.92) | $-$**25.73 (0.39)** | **98.72 (4.42)** |

We visualize optimization performance on these three objective functions in Fig. 4. Our proposed policy MPD is consistently competitive against both GIBO and TuRBO. Most notably, in the cosmological constant learning problem, MPD was able to make significant progress immediately and ultimately outperforms its closest spiritual competitor GIBO.

## 4.4 Ablation study

We now present results from various ablation studies to offer insight into the components of our method MPD and its hyperparameters, specifically the descent probability threshold $p_*$ (65% as the default) and the step size $\delta$ (0.001 as the default), as described in Sect. 3.

First, one may reasonably ask which of the two novel components of MPD – either the learning phase that seeks to maximize expected posterior descent probability, or the update phase that moves in the most probable descent direction – is responsible for the performance improvement compared to GIBO. We address this question by comparing the performance of MPD against two variants: (1) trace + **MPD**, which learns about the gradient by minimizing the trace of the posterior covariance matrix and moves in the most probable descent direction, and (2) **MPD + expected gradient**, which uses our scheme for identifying the most probable descent direction, then moves in the direction of the (negative) expected gradient. The second section of Tab. 1 shows the average terminal objective values of these MPD variants on three tasks that MPD outperforms GIBO: Swimmer-v1, cosmological constant learning, and rover trajectory planning. We observe that swapping out either component of MPD does not consistently improve from GIBO as much as MPD does. This indicates that the two components of our MPD algorithm work in tandem and both are needed to successfully realize our local BO scheme.

In particular, the components of our method are coupled: because the expected gradient and the most probable descent direction are not the same in general, spending evaluation budget to learn about one and then using the other to move may not work well. GIBO's acquisition function minimizes the trace of the posterior covariance and therefore aims to make the expected gradient estimate more

accurate, but it is unclear whether it will necessarily estimate the most probable descent direction accurately. On the other hand, our acquisition function focuses on the one-step maximum descent probability directly. GIBO's "moving" policy, moving in the direction of the (negative) expected gradient (which may not be the most probable descent direction), may not necessarily benefit from having a descent direction with a high descent probability (which could point in a different direction), and is therefore incompatible with our acquisition function.

We also tested MPD with three other values for the minimum descent probability threshold $p_* \in \{50\%, 85\%, 95\%\}$ (described in Sect. 3). The first variant with $p_* = 50\%$ is less conservative when moving to a new location than our default policy with $p_* = 65\%$, while the other two variants are more conservative. In the third section of Tab. 1, we observe that the more conservative variants of MPD are not as competitive. For example, MPD($p_* = 85\%$) sees a drop in performance on the Swimmer task, while MPD($p_* = 95\%$) fails to make significant progress altogether. Interestingly, while the less conservative policy with $p_* = 50\%$ also does not perform as well on the two Mujoco tasks, we do observe an increase in performance in the rover trajectory planning problem. From our experiments, we find that this rover objective function is piecewise linear within most of its domain, making finding a descent direction "easier" and allowing a lower value of $p_*$ to perform better.

The interpretation of the threshold $p_*$ is quite natural: it sets a threshold of the minimum probability that we would make progress by moving to a new location. Intuitively, this hyperparameter trades off robustness versus optimism, with higher thresholds spending more budget before moving, but being more confident in their moves. While $p_* = 65\%$ performs well in our experiments, a user can set their own threshold depending on their use case. As observed with the rover trajectory planning problem, if there are structures within the objective function that make it "easy" to find a descent direction, MPD may benefit from a lower threshold. We might also consider dynamically setting the value of $p_*$ based on recent optimization progress – that is, we might increase $p_*$ if we believe that we are approaching a local optimum and therefore that finding a promising descent direction is becoming more challenging.

Finally, the lower section of Tab. 1 shows the performance of the variants of MPD with two additional step sizes, $10^{-4}$ and $10^{-2}$. We observe that MPD with $\delta = 10^{-2}$ occasionally fails to perform better than GIBO, illustrating the potentially detrimental effect of a step size that is too large. This step size parameter $\delta$ balances between faster convergence and taking steps that are too large, analogous to gradient descent, and may even be problem dependent. It would be additionally interesting to analyze whether there are good "rules of thumb" for setting $\delta$ based on the length scale of the GP, as smoother functions can likely support larger step sizes.

## 5   Conclusions

We develop a principled local Bayesian optimization framework that revolves around maximization of the probability of descending on the objective function. This novel scheme is realized with (1) an update rule that iteratively moves from the current location in the direction of maximum descent probability, and (2) a mathematically elegant, computationally convenient acquisition function that aims to maximize the probability of descent prior to our next move. Our extensive experiments show that our policy outperforms natural baselines on a wide range of applications.

(Local) Bayesian optimization has seen a wide range of applications across science, engineering, and beyond; an extensive annotated bibliography of these applications was compiled by Garnett [6] [appendix D]. However, it is possible to leverage BO for nefarious purposes as well; a concrete example is constructing adversarial attacks on machine learning models [22, 24]. Further, BO requires human expertise and ethical considerations in many important applications, and fully automated optimization systems may run the risk of perpetuating misaligned goals in machine learning. The authors judge the potential positive impacts on society resulting from better methods for local optimization to outweigh the potential negative impacts.

## Acknowledgments and Disclosure of Funding

We thank Natalie Maus for her contribution to the initial stage of this work and the anonymous reviewers for their feedback during the review stage. QN and RG were supported by the National Science Foundation (NSF) under award numbers OAC–1940224, IIS–1845434, and OAC-2118201. KW and JRG were supported by NSF award number IIS-2145644.

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
