## A  Hyperparameters

We follow Müller et al. [18] and only train our GPs on the last $N_{\max}$ data points collected in the BO loop (see §3.3 of Müller et al. [18]). For the synthetic and reinforcement learning objective functions, we use the default settings provided by Müller et al. [18] (Appendix A.8). We report the hyperparameters used in the remaining three objectives in Tab. 2.

Table 2: Hyperparameters and hyperpriors for numerical experiments.

|  | lengthscales | outputscale | noise standard deviation | $M$ | $N_{\max}$ |
|---|---|---|---|---|---|
| Map fitting | $\mathcal{N}(2,1)$ | $\mathcal{N}(5,1)$ | 0.01 | 1 | 512 |
| Cosmo. constant | $\mathcal{U}(0.05,20)$ | $\mathcal{U}(0.05,20)$ | 0.01 | 1 | 32 |
| Rover trajectory | $\mathcal{N}(9,1)$ | $\mathcal{N}(5,1)$ | 0.01 | 1 | 32 |

## B  Further discussion on experiment results

As shown in Sect. 4, our method MPD in outperformed by other baselines on the Hopper task, in particular the GIBO method of Müller et al. [18]. Here, we present the findings of our investigation into this difference in performance.

First, we note that the GIBO authors tuned numerous algorithmic hyperparameters on a task-specific basis, including hyper-hyperparameters of hyperpriors, parameters of the "inner loop" optimization of SGD, etc. (see Tab. 3 in Appx. A.7 of Müller et al. [18]). We adopted all these hyperparameters for our experiments, artificially boosting GIBO's performance vs. that of MPD. That said, we do believe MPD's performance is fairly robust to most of these hyperparameters.

We point out one hyperparameter of relevance to Hopper alone among the RL tasks – the use of the state normalization scheme described by Müller et al. [18] in their §3.3. This is enabled only for Hopper, and in fact its use was deemed to be critical success in their ablation study (§4.4). On its face, the scheme is fairly simple – we normalize states according to a running average/standard deviation of each coordinate visited throughout optimization. However, we believe this normalization scheme has unexpected and unintended interactions with both the GIBO and MPD algorithms. Namely, both algorithms proceed by alternating between two distinct behaviors: an inner "learning" policy (lines 6–10 of Alg. 1), which evaluates the objective around a point $x$ to learn about the gradient $\nabla f(x)$, and an outer "moving" policy (lines 11–14 of Alg. 1), which uses this information to progress from $x$ to the next point in the domain to evaluate, $x \to x'$. Here (and in the GIBO codebase), state normalization was turned on for both learning and moving.

We argue that the running averages used for state normalization should only be based on the trajectory of the outer loop rather than additionally on the exploration we make along the way in the inner loop. Systematic differences in the behavior of the learning policy (say, generally sampling closer or farther away from $x$) will lead to systematic differences in the state normalization behavior. This will even affect the function values observed by outer loop policy even if the exact same trajectory is followed. Fig. 5 illustrates this behavior in a very simple setting. We show a sequence of Hopper function evaluations along a grid spanning a 1d linear subspace of the Hopper domain. Think of moving from point to point the grid as the trajectory of the "moving" policy. The only difference between these plots is that the function is evaluated 5 times at each location on the left and 20 times at each location on the right (think of re-sampling as the "learning" policy). The distribution of function values is dramatically different along the trajectory due to systematic (in this case, trivial) differences in the learning policy.

We aim to control for the effect of state normalization in our comparison between GIBO and MPD. A simple way to do this is to disable updates to the state normalization constants in the inner "learning" loop (evaluating the objective around the current location to learn about the gradient). That is, we only allow state normalization to be defined by the trajectory generated by the outer loop, where we move to new locations. We reran both algorithms and report the results in the second row of Tab. 3, where we see a drop from the original performance with state normalization updates always enabled (first row). We also observe statistically comparable performance between the two algorithms.

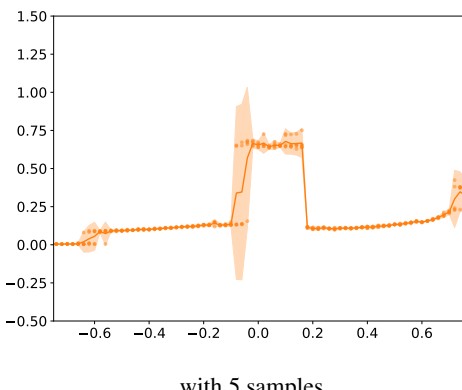
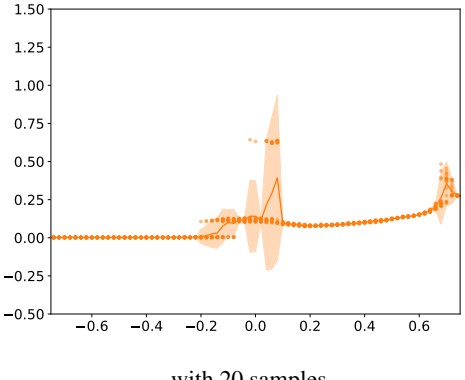

| with 5 samples | with 20 samples |
| :---: | :---: |

Figure 5: The values returned by the Hopper-v1 objective function with Müller et al. [18]'s state normalization scheme turned on along a grid spanning a 1d linear subspace of the Hopper domain. Each point is evaluated 5 times on the **left** and 20 times on the **right**. The orange dots show the values observed from the objective function, and the lines and shaded regions show the empirical mean and standard deviation. With the state normalization scheme, the trends observed in the objective function differ dramatically simply due to evaluating a different number of times.

Table 3: Average terminal optimized rewards on Hopper-v1 for different versions of GIBO and MPD.

| | GIBO | MPD |
| :---: | :---: | :---: |
| original (as reported in Sect. 4) | 2827.96 (273.02) | 2199.48 (337.44) |
| no updates to the state norm. constants | 1032.97 (375.51) | 1398.35 (307.29) |
| fixed state norm. from initial random exploration | 2100.63 (405.11) | 2086.74 (315.01) |
| without state norm. | 415.56 (64.82) | 381.50 (78.92) |

As another way of comparing GIBO and MPD while keeping the effect of state normalization fixed, we adopt the common practice in reinforcement learning (RL) in which we run an initial exploration phase (of 1000 random queries to the objective) to initialize the state normalization constants, and then run each algorithm with those constants fixed. This way, GIBO and MPD will always share the same state normalization scheme. The results of this setup are in the third row of Tab. 3, where we again observe statistically comparable performance. Given the apparent interaction between using state normalization and other hyperparameters considered, it' is plausible that the performance of both methods in the third row could be improved substantially by further tuning and engineering. Finally, we see that just as Müller et al. [18] reported in the ablation study in their §4.4, the performance of MPD suffers without state normalization (last row of Tab. 3). Once again, the two algorithms are statistically comparable.

Overall, we highlight the complex role state normalization plays in the Hopper experiments. While it is not clear why this state normalization scheme works so well for GIBO on this particular problem, we note that it only does with the various tuned hyperparameters that are specific to Hopper (for example, state normalization was not used in other RL problems). All of this engineering that has a significant impact on the final performance of these methods on Hopper suggests a significant degree of brittleness that is (a) undesirable in practice, and (b) not seen on even the other RL tasks we consider. This suggests that neither GIBO nor MPD may be the most robust optimization routine to use on Hopper specifically.

## C  Code and licenses

We use GPyTorch and BoTorch to extend GIBO, which is under the MIT license, and implement MPD. Implementation of objective functions used is curated from authors of respective publications,

as stated in Sect. 4. No identifiable information or offensive content is included in the data. Code implementation is included in the supplemental material and will be released under the MIT license.