# OpenReview forum: "Local Bayesian optimization via maximizing probability of descent"
_NeurIPS.cc/2022/Conference — NeurIPS 2022 Accept_

### Official Review · Reviewer_ohQU · 2022-07-03

**Rating:** 7
**Confidence:** 4
**Soundness:** 3 good
**Presentation:** 4 excellent
**Contribution:** 3 good

**Summary:**

The paper aims to improve Bayesian optimization in high-dimensional search spaces by proposing a new technique for local optimization. The authors build on prior work on local policy search with bayesian optimization (GIBO) by 1) showing that the expected value of the gradient may not coincide with the maximum probability of descent, and 2) proposing an optimization scheme which directly optimizes the probability of descent. In experiments, the approach tends to outperform GIBO as well as other more complicated baselines proposed in the literature.

**Questions:**

1. Do the authors have any intuition as to why swapping either of the components of MPD does not consistently improve over GIBO in the ablation study? Would we not have expected either one of the two components in isolation to already be an improvement? This may or may not be the case, but I would have liked the authors to give more intuition on this result.
2. Have the authors considered additional ablation studies to quantify the advantages of MPD's local search when plugged as the local routine in McLeod et al. or Wang et al.? While not strictly within the scope of the paper, demonstrating the benefits of MPD as a general technique that can boost existing local optimization method would step up the impact of the paper.
3. [key question] How did the authors come up with the default policy using p∗ = 65%? Since the ablation studies show that the results are to some extent sensitive to this choice, it is important to understand if 1) this is a general purpose default recommendation, and 2) how the recommendation was obtained (i.e., whether it was obtained on a given validation set before comparing to the baselines in Table 1).
4. Have the authors also run ablation studies on the step size, which was set to δ = 0.001? Any insights on the impact of larger and lower step size values?
5. Minor: Some of the compared baselines in Figure 2 do not seem to have fully "converged" (e.g., ARS on the 100D objective). If run for more queries, do all methods get to the same optimum? Do we observe any changes in ranking? While the used budget is arguably already very high, this would be interesting to confirm that on top of being the more query-efficient approach, MPD also tends to consistently converge to th best solution in the long run.
6. Do the authors have any insights into what makes 33D Hopper so different from the other problems that lead MPD to be subpar compared to all three baselines? This is surprising as for very high dimensional spaces, the advantage of MPD was more pronounced on the synthetic functions.
7. The evaluation considers number of queries on the x axis. What's the computational cost of the MPD local optimization step compared to baselines such as GIBO? What would the results look like with wall-clock time no the x axis?

**Limitations:**

The authors have adequately discussed the potential negative societal impact of the work, which coincides with the risks of any BO and generally AutoML black-box system (i.e., it can be exploited for malicious purposes). I agree with the authors that the benefits of the proposal outweight these risks.

Beyond that, I believe the main limitations lie in the empirical section, due to some mixed results and the lack of clarity on the process that led to the choice of the method settings (probability of descent and delta). More details in the questions above. I am ready to increase my score if these questions are addressed.

**Strengths And Weaknesses:**

**Clarity**

The paper was a pleasure to read. Very well-written, easy to follow and self-contained, with enough background on both general and local Bayesian optimization to bring the reader up to speed. It also includes great visualizations (Figure 1) that help build intuition on the behavior of the proposed approach.

**Originality / Significance**

The problem of scaling up Bayesian optimization to high dimensional spaces is significant and the paper builds on the success of local optimization techniques. Showing that the most likely descent direction does not always coincide with the negative expected gradient is an interesting result, worth of value on its own even if unplugged from the resulting method.

While to some extent the work incrementally builds on GIBO, the results improving the local optimization routine are general and can be plugged in any existing approach relying on local optimization (e.g., McLeod et al., Wang et al.). As such, the impact of the method is broader as improved results could potentially be observed for each of these baselines. For this reason, I would have liked to see experiments validating the hypothesis that the proposal is in fact a useful subroutine to replace previous local optimization across the board. This would have further increased the significance and breadth of impact of the paper.

**Quality**

The paper's quality is overall high. The experimental evaluation is carefully carried out and includes ablation studies aiming at separating out the two key contributions of the proposed MPD approach (i.e., the learning phase or the update phase). From these results, it turns out that MPD does not perform consistently better against GIBO when one of the two new components is removed. This shows that both components need to be used in tandem to achieve improved results.

The proposal is theoretically grounded and the empirical evaluation is overall solid. The results show that the proposed approach tends to outperform baselines and particularly so as the dimensionality of the space grows on synthetic functions. While very promising and clearly outperforming baselines on these examples, the gap is not as pronounced on real-world objectives (Figure 3) and MPD is clearly worse on 33D Hopper. It is expected that not 100% of problems will show the benefits of MPD, but I would have liked the authors to elaborate and give intuition as to why there is such a clear drop in 33D Hopper.

Beyond that, I particularly appreciated the results on interesting real-world problems. This sets an example for work in the BO space to evaluate the proposed methods beyond standard problems, such as those from hyperparameter optimization.

---

> ### Author Response · Authors · 2022-08-02
> **Response to Reviewer ohQU**
>
> Thank you for your detailed reviews.
>
> **“Do the authors have any intuition as to why swapping either of the components of MPD does not consistently improve over GIBO in the ablation study? Would we not have expected either one of the two components in isolation to already be an improvement? This may or may not be the case, but I would have liked the authors to give more intuition on this result.”**
>
> We believe that the components of our method are coupled: because the gradient and the maximum probability descent direction are not in general the same, spending evaluation budget to learn about one and then using the other to move may not work well. GIBO’s acquisition function minimizes the trace of the posterior covariance and therefore aims to make the expected gradient estimate more accurate, but it is unclear whether it will necessarily yield likely descent. On the other hand, our acquisition function focuses on the one-step maximum descent probability directly. GIBO’s “moving” policy, moving in the direction of the (negative) expected gradient (which may not be the MPD), may not necessarily benefit from having a high MPD (which could point in a different direction), and is therefore compatible with our acquisition function.
>
> We will expand this discussion in the final version of the main text.
>
> **Plug in a larger optimization routine**
>
> This is indeed an interesting direction, especially since these local methods may significantly benefit from some global planning such as early stopping and restarting when we are certain that we have reached a local optimum. As future work, we plan to explore such “outer loop” schemes that involve several runs of a local optimizer like MPD.
>
> **How did the authors come up with the default policy using $p_∗ = 65\%$?**
>
> Please refer to our general response on this issue.
>
> **Ablation study on the step size $\delta$**
>
> This is a good suggestion, and we will include an additional ablation study over $\delta$. This parameter trades off between faster convergence and taking steps that are too large, analogous to gradient descent, and may even be problem dependent. It would be additionally interesting to analyze whether there are good “rules of thumb” for setting delta based on the lengthscale of the GP, as smoother functions can likely support larger values of $\delta$.
>
> **“Some of the compared baselines in Figure 2 do not seem to have fully converged..”**
>
> Thank you for the suggestion. We computed the average optimal function value in these synthetic objective functions and obtained the following numbers: $2.345$ for the 25-dimensional objectives, $2.599$ for the 50-dimensional objectives, and $2.175$ for the 100-dimensional objectives. While the values our method converges to are not exactly those global optima, they are fairly close. For example, for the 100-dimensional objectives, the normalized gap $(f^* - f) / f*$ the methods achieve is around $0.1$. We also found that given double the budget, ARS also converges to the same value.
>
> **“Do the authors have any insights into what makes 33D Hopper so different..”**
>
> Please refer to our general response on this issue.
>
> **“The evaluation considers the number of queries on the x axis…”**
>
> MPD is indeed more computationally expensive than GIBO, but MDP still runs quite efficiently in our experiments. For example, each run of GIBO making 1000 queries of the 200-dimensional Rover objective took roughly 20 minutes; for MPD, this took ~45 minutes. This is a difference of roughly 1.5 seconds/iteration in runtime. We also note that the main computational bottleneck of any Bayesian optimization algorithms in the real world is most likely going to be querying the objective function. Therefore, the total wall-clock time of a run of either approach would indeed be roughly determined by the number of queries made and not the ~1.5 seconds per iteration difference in the computation stage seen here.

---

> > ### Comment · Reviewer_ohQU · 2022-08-07
> > **Score increased**
> >
> > I'd like to thank the authors for the thorough responses, which addressed my main concern (particularly on the choice of the default policy value). I have increased my score to 7. I recommend the authors to include these discussions in the final version of the paper.

---

### Official Review · Reviewer_ogaD · 2022-07-11

**Rating:** 7
**Confidence:** 4
**Soundness:** 3 good
**Presentation:** 3 good
**Contribution:** 3 good

**Summary:**

This paper proposes a local Bayesian optimization strategy via maximizing a probability of descent.  Unlike the previous work (Muller et al. 2021), this work computes the most likely descent direction by considering the problem of the expected gradient.  More precisely, as described in the paper, the expected gradient is not necessarily the direction maximizing the probability of descent.  The proposed method solves this problem by computing the most likely descent direction.  Finally, the authors demonstrate that the proposed method works well in various experiments.

**Questions:**

I would like to ask the authors some questions.

1. In Line 88, a sentence, `Our work addresses this gap.` should be re-written.  Please clearly write the sentence by including what this gap is in the sentence.  I understand that it is not a final version, but please fix it in the final version.

1. In Equation 4, why is an acquisition function over a batch of potential query points?  I think that an acquisition function over a single query is tested (as in Algorithm 1).

1. Please add line numbers in Algorithm 1.

1. In experiments, why are some results fluctuated?  Why are not the best minimum (or maximum) values plotted?

**Limitations:**

I do not think that this work has any negative societal impacts and any specific limitations.

**Strengths And Weaknesses:**

## Strengths

+ It solves an interesting topic on local Bayesian optimization.

+ It suggests a novel method by raising the drawbacks of the existing method.

+ Experimental results do not always show the best results, which is very reasonable.  But please describe more thorough analyses on these results.

## Weaknesses

- I do not have specific weaknesses, but writing and presentation can be improved to clearly demonstrate the contributions of this work.

Please see the text box described below.

---

> ### Author Response · Authors · 2022-08-02
> **Response to Reviewer ogaD**
>
> Thank you for your comments.
>
> We will polish the sentence in Line 88 to make it clearer. The “gap” in Line 88 is referring to the fact that GIBO fails to account for the nuances in the belief of the gradient, since it uses the (negative) expected gradient directly as a descent direction, which may or may not be a particularly likely descent direction.
>
> We will add line numbers to Algorithm 1.
>
> **In experiments, why do some results fluctuate? Why are not the best minimum (or maximum) values plotted?**
>
> In the spirit of local optimization, the plots show the observed value _at the current location_ during optimization, i.e., we plot $y_t$ vs $t$. Since we move to $x_{t + 1}$ even if $f(x_{t + 1}) < f(x_t)$, the curve may not be monotonic. See also the discussion with Reviewer stFc on this point.

---

> > ### Comment · Reviewer_ogaD · 2022-08-05
> > **Thank you for your response!**
> >
> > Thank you for your response!
> >
> > I understood your paper better with your response.
> >
> > After reading the response and other reviewers' comments, I would like to maintain my score.
> >
> > Please consider and reflect all the comments in the final version.

---

### Official Review · Reviewer_stFc · 2022-07-11

**Rating:** 8
**Confidence:** 4
**Soundness:** 4 excellent
**Presentation:** 4 excellent
**Contribution:** 3 good

**Summary:**

This paper studies local Bayesian optimization in which a GP surrogate is used to give a model for the gradient at a point, an acquisition function is used to select points that are informative about the gradient, and then gradient steps are taken to select the next point for evaluation. Past work has used information gain about gradient for acquisition, and the direction of negative expected gradient for taking the gradient step. This paper builds on that work by showing that it is better to move in the direction that maximizes the probability of descent, which because of posterior uncertainty in the gradient, is not necessarily the negative expected gradient. It shows how this quantity can be computed, and then proposes an analytic acquisition function that targets the same quantity. Empirical results show strong performance relative to the work that this builds on, as well as other related work in this same problem space.

**Questions:**

When we observe the new value f(x_{t+1}), do we always accept the move from x_{t} to x_{t+1} or do we accept it only if f(x_{t+1}) < f(x_{t})? I imagine that observing x_{t+1} will update the gradient information at x_{t}, so even if we do not accept the step and we repeat the acquisition optimization around the same x_{t} as in the previous optimization, we should take different steps. It sounds from the paper that we are always accepting steps, but I wonder if that is the best strategy.

Is there anything about the 33D Hopper problem that explains why it performs worse?

Was the RBF kernel ARD?

The paper gives results on sensitivity to p* but only for more conservative choices; what about e.g. 0.5?

**Strengths And Weaknesses:**

I found this paper to be very interesting. It studies an important problem (high-dimensional BO). While it does build directly on top of existing work and so does not present an entirely new approach to the problem, it gives a very insightful analysis into the failure cases for that existing work and does a great job at motivating the new method. The paper is very clearly written and the work is of high quality.

The new method is surprisingly elegant, and is strongly motivated by a very clear mathematical analysis as well as a great illustration in Fig. 1.

The empirical results are very strong and compare to what I think are the most appropriate baselines. An ablation study is included, as are real-world problems.

Below I give a few questions that I had that I expect other readers may be interested in as well, though I consider these all to be minor issues.

---

> ### Author Response · Authors · 2022-08-02
> **Response to Reviewer stFc**
>
> We thank you for your comments.
>
> **Always accepting the move from $x_{t}$ to $x_{t+1}$ vs accepting only if $f(x_{t+1}) < f(x_{t})$**
>
> We do indeed always accept $x_{t + 1}$ in the reported results. Choosing not to move could indeed lead to a different maximum-descent direction in the next iteration, as you point out. However, we followed standard practice with, e.g., SGD methods and chose to always move. GIBO made the same decision.
>
> **33D Hopper**
>
> Please refer to our general response on this issue.
>
> **“Was the RBF kernel ARD?”**
>
> Yes, except for the Rover objective function, where we found the isotropic RBF kernel leads to better performance.
>
> **“...what about $p_*=0.5$?”**
>
> Please refer to our general response on this issue.

---

> > ### Comment · Reviewer_stFc · 2022-08-03
> > **Response**
> >
> > Thank you for answering these questions. Especially for the analysis of the Hopper problem which was very insightful.

---

### Official Review · Reviewer_Qcf8 · 2022-07-12

**Rating:** 6
**Confidence:** 4
**Soundness:** 2 fair
**Presentation:** 3 good
**Contribution:** 2 fair

**Summary:**

The paper develops a new approach for local Bayesian optimization by incorporating gradient into the Bayesian optimization as in the work by Muller et al [13] but instead of moving in the direction of the expected gradient, the paper proposes to identify the direction of most likely descent, and then move in that direction. To do so, the paper derives a closed-form solution for the direction of most likely descent at a given location in the input space under a GP belief about the objective function. A corresponding closed-form acquisition function that optimizes (an upper bound of) the one-step maximum descent probability is also developed to form the optimization scheme. Experiments on both synthetic and real-world tasks are conducted to demonstrate the performance of the proposed approach.

################################ AFTER REBUTTAL ###################################
Thanks authors for your detailed response. I understand more about the behaviours of the proposed approach and I do not have any concerns about the Hopper function anymore. I have been thinking again about the work, and I think maybe I'm indeed picky, so I decided to increase my score to 6.

**Questions:**

Apart from my comments in the Weakness section which the authors can response, I have the following additional questions:

•	How does the hyperparameter p^* affect the performance of the proposed approach?

•	What can be the reasons that the proposed approach performs badly for the objective function 33D Hopper?


**Limitations:**

The paper has some discussions on the societal impact of their work.

**Strengths And Weaknesses:**

Strengths: The paper is very well-written. All the concepts and the proposed approach are described clearly and easy to understand. The idea of incorporating the gradient information into the optimization process and move to the direction of most likely descent is sound and reasonable. Rigorous mathematical formulas are developed to derive the key components of the proposed approach (the direction of most likely descent and the acquisition function). The proposed approach has good performance in some objective functions.

Weaknesses: Some of the weaknesses of the paper are as follows:

•	Maybe I'm picky but I think the idea is not too novel as it feels like it just fixes a particular issue of an existing BO approach. I do appreciate all the analysis conducted to show the issue of the approach in Muller et al [13] and all the derivations to find the most likely descent direction and to compute the new acquisition function. However, I normally expect the key idea to be more novel, and if it's not too novel, I expect the experiment results to be very impressive. In this case, the experiment results are not really that impressive. The proposed approach does have some very good performance on some objective functions; however, it also performs on par with other baselines in multiple objective functions, and it performs worse for some objective functions.

•	The proposed approach is developed for the batch setting; however, only sequential setting (batch 1) is conducted in the experiments.

•	The paper seems to lack of the discussion about some benchmarks on Bayesian optimization with gradient, for example the work Bayesian Optimization with Gradients by Wu et al (NeurIPS 2017).

•	I expect more sensitivity analysis and discussions on some hyperparameters of the proposed approach such as p^*.

---

> ### Author Response · Authors · 2022-08-02
> **Response to Reviewer Qcf8**
>
> Thank you for your thoughtful feedback.
>
> **Related paper by Wu et al., 2017.**
>
> We note that the setting of this study is different, as the algorithm of Wu et al., 2017 assumes the ability to compute gradient of the objective function directly, whereas we only assume access to function evaluations without access to the gradient. With direct access to the gradient, identifying a descent direction becomes trivial, as we can simply take the gradient. Further, Wu, et al. considers global rather than location optimization. For these reasons, it is not fair to compare our algorithm with theirs. However, we will happily add discussion on Wu et al., 2017 to the related work section.
>
> **“I expect more sensitivity analysis…” and “How does the hyperparameter p affect the performance…”**
>
> We have presented the performance of different thresholds for $p_*$ in Table 4. We will include additional thresholds.
>
> **Performance on Hopper.**
>
> Please refer to our general response on this issue.

---

> > ### Comment · Reviewer_Qcf8 · 2022-08-06
> > **Response to the authors' response**
> >
> > Dear authors,
> >
> > I just want to acknowlege that I've read your response, and still wait for the results of the Hopper function.

---

### Author Response · Authors · 2022-08-02
**General responses to all reviewers**

We thank all reviewers for their feedback on our paper.  We would like to respond to some common questions here.

**Experiments on Hopper**

As pointed out by the reviewers, MPD is outperformed by other baselines on the Hopper task, in particular the GIBO method of Müller, et al. We have investigated this issue thoroughly in the past week and believe we have identified the culprit.

First, we note that the GIBO authors tuned numerous algorithmic hyperparameters on a task-specific basis, including hyper-hyperparameters of hyperpriors, parameters of the "inner loop" optimization of SGD, etc. (see Table 3 in appendix A.7 of their supplement). We adopted these hyperparameters wholesale for our experiments, artificially boosting GIBO's performance vs that of MPD. That said, we do believe MPD's performance is fairly robust to most of these hyperparameters.

However, we have isolated one hyperparameter of relevance to Hopper alone among the RL tasks -- the use of the state normalization scheme described by Müller, et al. in their §3.3. This is enabled only for Hopper, and in fact its use was deemed to be critical success in their ablation study (§4.4). On its face, the scheme is fairly simple -- we normalize states according to a running average/standard deviation of each coordinate visited throughout optimization.

However, we believe this normalization scheme has unexpected and unintended interactions with both the GIBO and MPD algorithms. Namely, both algorithms proceed by alternating between two distinct behaviors: an inner "learning" policy, which evaluates the objective around a point x to learn about the gradient ∇f(x), and an outer "moving" policy, which uses this information to progress from x to the next point in the domain to evaluate, x ↦ x'. _Here (and in the GIBO codebase) state normalization was turned on for both learning and moving._

We argue that the running averages used for state normalization should only be based on the trajectory of the outer loop rather than additionally on the exploration we make along the way in the inner loop. Systematic differences in the behavior of the learning policy (say, generally sampling closer or farther away from x) will lead to systematic differences in the state normalization behavior. _This will even effect the function values observed by outer loop policy even if the exact same trajectory is followed._

We have updated our supplement to include a new figure (Figure 5) illustrating this behavior in a very simple setting. We show a sequence of Hopper function evaluations along a grid spanning a 1d linear subspace of the Hopper domain. Think of moving from point to point the grid as the trajectory of the "moving" policy. The only difference between these plots is that the function is evaluated 5 times at each location on the left and 20 times at each location on the right (think of resampling as the "learning" policy). The distribution of function values is dramatically different along the trajectory due to systematic (in this case, trivial) differences in the learning policy.

Of course, there is a simple fix for this -- we simply turn off state normalization updates in the inner loop while keeping them on during the outer loop. We are making this change now and will post updated results in the coming week.

---

> ### Author Response · Authors · 2022-08-02
> **General responses to reviewers (cont.)**
>
> **Clarification on the probability threshold $p_*$**
>
> Several questions arose on this topic.
>
> Several reviewers have asked about our choice of the $p_*$ threshold, and specifically why we choose conservative thresholds—e.g., $p_* = 65\%$. We would like to first note that it is always possible to find a direction with at least $50\%$ descent probability, as if the descent probability of a given direction is less than $50\%$, then the descent probability in the opposite direction is more than $50\%$. In fact, $p_* = 50\%$ if and only if $\mathbb{E}[g] = 0$, i.e., the expected gradient is zero. Thus setting $p_* = 50\%$ will lead us to a critical point of the posterior mean.
>
> That said, we ran this less coservative version of MPD with $p_* = 50\%$ on two objectives from our experiments, Swimmer from Mujoco and cosmological constant learning, and observed a slight decrease in performance compared to $p_* = 65\%$. Intuitively, the $p_*$ threshold trades off robustness versus optimism, with higher thresholds spending more budget before moving, but being more confident in their moves.
>
> Average terminal optimized rewards on selected objective functions for two versions of MPD.
>
> |  | Swimmer (maximization) | Cosmo. constant (maximization) |
> |--|--|--|
> | MPD($p_* = 65\%$) | 360.81 | -23.97 |
> | MPD(p* = 50%) | 342.36 | -24.29 |
>
> Some reviewers ask how the choice for $p_*$ in 65%, 85%, 95% was made. We simply rounded the upper-tail probabilities of a standard normal distribution corresponding to $z = 1$, $1.5$, and $2$ standard deviations to the “nice” values of $65\%$, $85\%$, and $95\%$, respectively.
>
> We also note that the interpretation of this hyperparameter is quite natural: it sets a threshold of the minimum probability that we would make progress by moving to a new location. While $65\%$ performs well in our experiments, a user can set their own $p_*$ depending on their use case. For example, if there are structures within the objective function that make it “easy” to find a descent direction, MPD may benefit from a lower threshold. We might also consider dynamically setting the value of $p_*$ depending on the recent optimization progress -- that is, we might increase $p_*$ if we believe that we are approaching a local optimum and therefore that finding a promising descent direction is becoming more challenging.
>
> We will add this discussion to the final version of the paper.
>
> **Batch query in Eq 4 v.s. Single query in Algorithm 1.**
>
> When deriving the acquisition function, we wanted to make it as general as possible and thus derived the batch version in full generality. In the experiments, we use a single query in order to make the comparison with GIBO consistent, as GIBO uses single queries.

---

> ### Author Response · Authors · 2022-08-09
> **More comments on Hopper**
>
> We thank the reviewers for their patience.
>
> We further our discussion on the Hopper experiments here. We earlier argued that the online state normalization scheme employed by Müller et al. might have unintended interactions with both GIBO and MPD. Specifically, systematic differences in the behavior of the algorithms lead to systematic differences in the state normalization behavior, and we wish to control for the effect of state normalization in our comparison between GIBO and MPD.
>
> As noted in our previous comment, a simple way to do this is to disable updates to the state normalization constants in the inner “learning” loop (evaluating the objective around the current location to learn about the gradient). That is, we only allow state normalization to be defined by the trajectory generated by the outer loop, where we move to new locations. We reran both algorithms and report the results in the second row of Table 1, where we see a drop from the original performance with state normalization updates always enabled (first row). We also observe statistically comparable performance between the two algorithms.
>
> Table 1: Average terminal optimized rewards on Hopper for different versions of GIBO and MPD.
>
> |  | GIBO | MPD |
> |---|:---:|:---:|
> | Original (as reported in the paper) | 2827.96 (±273.02) | 2199.48 (±337.44) |
> | Inner loop does not update the state normalization constants | 1032.97 (±375.51) | 1398.35 (±307.29) |
> | Fixed state normalization from initial random exploration | 2100.63 (±405.11) | 2086.74 (±315.01) |
> | Without state normalization | 415.56 (±64.82) | 381.50 (±78.92) |
>
> As another way of comparing GIBO and MPD while keeping the effect of state normalization fixed, we adopt the common practice in reinforcement learning (RL) in which we run an initial exploration phase (of 1000 random queries to the objective) to initialize the state normalization constants, and then run each algorithm with those constants fixed. This way, GIBO and MPD will always share the same state normalization scheme. The results of this setup are in the third row of Table 1, where we again observe statistically comparable performance. Given the apparent interaction between using state normalization and other hyperparameters considered, it’s plausible that the performance of both methods in the third row could be improved substantially by further tuning and engineering.
>
> Finally, we see that just as Müller et al. reported in the ablation study in their §4.4, the performance of MPD suffers without state normalization (last row of Table 1). Once again, the two algorithms are statistically comparable.
>
> Overall, we highlight the complex role state normalization plays in the Hopper experiments. While it is not clear why this state normalization scheme works so well for GIBO on this particular problem, we note that it only does with the various tuned hyperparameters that are specific to Hopper (for example, state normalization was not used in other RL problems). All of this engineering that has a significant impact on the final performance of these methods on Hopper suggests a significant degree of brittleness that is (a) undesirable in practice, and (b) not seen on even the other RL tasks we consider. All of this suggests that neither GIBO nor MPD may be the most robust optimization routine to use on Hopper specifically. We will add these discussions on Hopper to the final version of the paper.

---

### Meta-Review · Area_Chair_rxsB · 2022-08-24

**Recommendation:** Accept
**Confidence:** Certain

**Metareview:**

All reviewers are positive and agree that the paper should be accepted. The primary question raised about the poor performance on Hopper was adequately addressed by the author response. Please integrate the changes and reviewer suggestions around clarity into the final paper.

**Award:**

No

---

### Decision · Program_Chairs · 2022-09-14

Accept